# Trajectory of Diastolic Function after Heart Transplantation as Assessed by Left Atrial Deformation Analysis

**DOI:** 10.3390/diagnostics14111136

**Published:** 2024-05-29

**Authors:** Borbála Edvi, Alexandra Assabiny, Tímea Teszák, Máté Tolvaj, Alexandra Fábián, István Hartyánszky, Miklós Pólos, Bálint Károly Lakatos, Hajnalka Vágó, Balázs Sax, Béla Merkely, Attila Kovács

**Affiliations:** 1Heart and Vascular Center, Semmelweis University, H-1122 Budapest, Hungaryteszaktimea@gmail.com (T.T.);; 2Department of Experimental Cardiology and Surgical Techniques, Semmelweis University, H-1122 Budapest, Hungary

**Keywords:** heart transplantation, diastolic function, left atrium, longitudinal strain, heart failure

## Abstract

Diastolic dysfunction (DD) is a prevalent and clinically significant complication after heart transplantation (HTX). We aimed to characterize the diastolic function of HTX recipients with both short-term and long-term follow-ups by applying left atrial (LA) deformation analysis. We consecutively enrolled and followed up with 33 HTX patients. Three assessments were performed one month, 3–5 months, and 3–5 years after surgery. Beyond conventional echocardiographic measurements, apical four-chamber views optimized for speckle tracking analysis were acquired and post-processed by dedicated software solutions (TomTec AutoStrain LA and LV). Left atrial phasic functions were characterized by reservoir, conduit, and contraction strains. We categorized diastolic function according to current guidelines (normal diastolic function, indeterminate, DD). At the first assessment, nine (27%) patients were in the DD category, and eleven (33%) were indeterminate. At the second assessment, only one patient (3%) remained in the DD category and six (18%) were indeterminate. At the third assessment, 100% of patients were categorized as having normal diastolic function. LA reservoir strain gradually increased over time. LA contraction strain significantly improved from the second to the third assessment. We found a correlation between the LA reservoir strain and NT-proBNP (r = 0.40, *p* < 0.05). DD is prevalent immediately after HTX but rare until the end of the first postoperative quarter. Speckle tracking analysis enables the characterization of LA phasic functions that may reflect both short- and long-term changes in diastolic function and correlate with NT-proBNP.

## 1. Introduction

Diastolic dysfunction (DD) is one of the complications that can occur after cardiac allograft, but its actual frequency, progression, or regression dynamics have not been well described [1]. Many transplant patients have DD soon after surgery, but it becomes less frequent in the first year [2]. However, persistent and invasively confirmed DD, especially if coupled with decreased cardiac output, predicts mortality [1]. Thus, there is a need for a continued, comprehensive, and non-invasive assessment allowing for the sensitive detection and monitoring of DD in heart transplant (HTX) recipients.

Echocardiography is the prime modality for detecting DD. Current guidelines have introduced complex pathways for a definite diagnosis; moreover, after HTX, altered anatomical and hemodynamic factors introduce further limitations for conventional echocardiographic parameters [3]. This population lacks a standardized approach and specific normative values. Speckle tracking echocardiography-derived left atrial (LA) longitudinal strain adds diagnostic and prognostic value to numerous cardiac pathologies. Therefore, it is a candidate parameter to be included in future recommendations [4]. However, while LA strain mainly reflects secondary mechanisms in the intact heart, leading to increased filling pressures, the function of the cardiac allograft is affected by several factors, including the donor, recipient, and surgical technique. Yet, its changes after HTX have not been fully investigated.

Accordingly, we aimed to characterize the diastolic function of heart transplant recipients with both short-term and long-term follow-ups by applying LA deformation analysis.

## 2. Methods

We consecutively enrolled and followed up 33 patients transplanted to our center from January 2018 to January 2019. The first echocardiographic assessment was performed one month after HTX in patients discharged from the intensive care unit and presented with a stable hemodynamic condition. The second echocardiographic assessment was performed 3–5 months after HTX aligned with scheduled clinical visits. The third echocardiographic assessment was performed 3–5 years after HTX, again aligned with scheduled clinical visits. As for exclusion criteria, patients with ISHLT grade ≥II rejections diagnosed by concurrent endomyocardial biopsy were excluded from the analysis, and investigated again later (no such assessment occurred). Patients were provided written informed consent prior to the study procedures.

Echocardiographic acquisitions were performed on EPIQ 7 (Philips, Andover, MA, USA) or Vivid E95 (GE Vingmed Ultrasound, Horten, Norway) ultrasound systems. Standard acquisition protocol consisting of loops from parasternal, apical, and subxiphoid views was used according to current guidelines [5]. Left ventricular (LV) internal diameters, wall thicknesses, relative wall thickness, and mass; LA 2D end-systolic volume; mitral inflow velocities such as early (E) and late diastolic (A) peak velocities, their ratio, and E wave deceleration time; systolic (s′), early diastolic (e′), and atrial (a′) velocities of the mitral lateral and septal annulus; and average E/e′ were measured accordingly. LV end-diastolic (EDV), end-systolic (ESV) volumes, and ejection fraction (EF) were quantified using the biplane Simpson’s method. Right ventricular (RV) basal diameter and M-mode-derived tricuspid annular plane systolic excursion (TAPSE) were measured from RV-focused apical four-chamber views. Valvular heart diseases were assessed according to guideline recommendations. Tricuspid regurgitation jet peak velocity was noted as a surrogate for peak pulmonary arterial pressure.

Based on current guideline recommendations, we categorized each patient at each time point into normal diastolic function, indeterminate, and DD subgroups [3]. We used the following pathological cutoffs: average E/e′ > 14; septal e′ velocity < 7 cm/s or lateral e′ velocity < 10 cm/s; tricuspid regurgitation peak velocity >2.8 m/s; and LA volume index >34 mL/m^2^. Diastolic function is normal if more than half of the available variables do not meet the cut-off values for identifying DD. DD is present if more than half of the available parameters meet these cut-off values. The category is indeterminate if half of the parameters do not meet the cut-off values [3].

We analyzed LV and LA deformation using a vendor-independent software package (AutoStrain LV and AutoStrain LA, TomTec Imaging Systems, Unterschleißheim, Germany) on the apical four-chamber view. We did not analyze apical two-chamber and long-axis views to avoid losing patients due to image quality or availability. We measured LV global longitudinal strain (GLS—average of six LV segments), LA reservoir, conduit, and contraction strains referring to the phasic LA functions. The software detected the apical four-chamber view and traced the LV and LA endocardial borders frame-by-frame as they changed in length along the cardiac cycle. In some cases, poor ECG or 2D echocardiographic image quality required manual adjustment of cardiac cycle events or the endocardial border, respectively. The ventricular end-diastole was the reference time point for LA deformation analysis.

Normal distribution of variables was tested using the Shapiro–Wilk test. As the majority of the variables were non-normally distributed, continuous data were presented as median and interquartile ranges (first quartile; third quartile), while categorical data were presented as counts and percentages (% of the total population). The Sankey diagram was constructed using SankeyMATIC (https://sankeymatic.com, accessed on 23 May 2024) to visualize the volume of reclassified patients over 3 time points. To compare variables across 3 time points, repeated measures ANOVA was used. Time point pairs were compared with the least significant difference test in normal distribution. For non-normal distribution, the Friedman test was used, followed by the Wilcoxon test with Bonferroni adjustment. Spearman’s rank correlation test assessed the correlation between the variables and *p*-values < 0.05 were considered statistically significant.

## 3. Results

Basic demographic, hemodynamic, and clinical characteristic data of the recipients, along with donor age and sex, are summarized in Table 1. The median age of the HTX patients at the first echocardiographic assessment was 52 years and nine (27%) were female. Nonischemic dilated cardiomyopathy (49%) was the dominant etiological factor for the HTX indication. Nine (27%) patients had ISHLT grade ≥ II rejection episodes until the third assessment. Five patients (15%) died between the second and third echocardiographic assessment; thus, the comparisons presented refer to only 28 patients (Table 2).

Conventional and speckle tracking echocardiographic parameters at three time points are summarized in Table 2. Interventricular septal thickness decreased from the second to the third assessment. Other LV diameters and volumes did not change. While LVEF decreased at the third assessment, GLS showed numerical improvement (statistically non-significant), suggesting hemodynamic and morphological changes but not decreasing LV systolic function. The prevalence of more than moderate TR significantly decreased from the first assessment to the second and third (18 [54%] vs. 10 [30%] vs. 10 [30%] patients), along with an increase in RV function as assessed by TAPSE. TR Vmax values decreased from the second assessment to the third.

According to guideline recommendations, we categorized each patient at each time point into normal diastolic function, indeterminate, and DD subgroups [3]. While at the first assessment (1 month after HTX), nine (27%) patients were in the DD category, and eleven (33%) were indeterminate. At the second assessment (3–5 months after HTX), only one patient (3%) remained in the DD category and six (18%) in indeterminate (Figure 1). One hundred percent of patients were categorized as having normal diastolic function at the third assessment (3–5 years after HTX). Four of the five deceased patients had normal diastolic function at the second assessment, and the fifth was in the indeterminate category (Figure 2). All nine patients with ISHLT grade ≥ II rejection episodes had a normal diastolic function before the episode at the second assessment. Eight patients remained in the normal function category after it, while one patient died before the third assessment. 

Mitral inflow E velocity decreased in the second assessment. Similarly, tissue Doppler imaging-derived lateral e′ velocity improved from the first assessment to the second, along with E/e′, but did not change in the third assessment. By contrast, LA volume decreased only in the third assessment. LA reservoir strain gradually increased over time and was statistically significant between the first and second assessments. Interestingly, the LA contraction strain significantly improved from the second to third assessment (Table 2 and Figure 3). 

By pooling all time point values, we found a correlation between the LA reservoir strain and NT-proBNP (r = 0.40, *p* < 0.05) as well as between the temporal changes of the LA reservoir strain and changes in NT-proBNP (r = −0.38, *p* < 0.05).

## 4. Discussion

Our study results can be summarized as follows: (1) Diastolic dysfunction, as per current echocardiographic guideline recommendations, is prevalent immediately after HTX but becomes rare until the end of the first postoperative quarter, completely disappearing in long-term follow-up. (2) Conventional echocardiographic parameters of diastolic function show varying patterns of temporal changes. (3) Speckle tracking-based deformation analysis for LA phasic functions may reflect both short- and long-term changes in diastolic function and correlate with NT-proBNP.

Many factors affect transplanted hearts’ diastolic function, making DD assessment and detection cumbersome. One factor is denervation, resulting in sinus tachycardia and subsequent fusion of the E and A waves. There is also a difference between biatrial and bicaval surgical techniques (the latter was used in all cases in our patients). The mitral inflow resembles a restrictive pattern in most cases, especially in the early postoperative period, irrespective of rejection status [3]. Similarly, tissue Doppler imaging-derived parameters can provide inconsistent results as the insonation angle heavily influences them. The literature suggests they are reduced over one year compared to a healthy population [2]. 

Speckle tracking echocardiography enables mechanical characterization of all phasic functions of the LA, namely reservoir, conduit, and booster pump. As a non-Doppler technique, it is also less dependent on anatomical variations and insonation angle. Similarly to LV and RV deformation parameters, LA reservoir strain may have a role in detecting acute cellular rejections after HTX [6,7,8]. Zhu et al. demonstrated that LA functions by longitudinal strain are significantly different (impaired) compared with controls, irrespective of surgical technique [9]. Not surprisingly, the reservoir strain is correlated with recipient age, LA volume, and LV function. In line with these results, we measured significantly lower LA strain values compared to available normative data [10]; however, we also showed that LA functions improve over time. Reservoir strain improves early along with, for example, E/e′, and its absolute values and changes correlate with NT-proBNP. The latter finding is remarkable as this phenomenon may reflect that, similar to NT-proBNP, reservoir strain can be a marker of “overall” cardiac health by integrating LV systolic and diastolic function, filling pressures, and structural remodeling. Although the association of absolute NT-proBNP levels with acute cellular rejection is weak, temporal changes provide important clinical information [11]. The neurohormone’s temporal changes were also associated with temporal changes in the LA reservoir strain, suggesting a pathophysiological link between the two biomarkers and justifying Rodriguez-Diego’s findings about reservoir strains’ usefulness in detecting rejection episodes [6]. 

Importantly, however, LA contraction strain improves in the long term. The phenomenon may be more associated with structural remodeling, as seen in the significant decrease in LAVi at our third assessment. Thus, future studies with higher case numbers and more frequent assessments should be directed toward understanding the determining factors of LA structural and functional remodeling after HTX. In addition, they should establish the added value of this temporal remodeling pattern in detecting DD and rejection episodes.

## 5. Limitations

Some limitations of our study must be acknowledged. Firstly, the patient number is relatively low; however, the value of the two (short- and long-term) follow-up examinations should be highlighted. Nevertheless, we could not assess the prognostic value of DD due to limited case numbers. “Gold standard” measures of elevated filling pressures or magnetic resonance imaging-derived tissue characterization were unavailable. Although the LA strain is a promising new parameter for assessing cardiac allografts, typical cut-off values from literature data are not suitable for HTX patients. For these reasons, when examining the diastolic function of a transplanted heart, it is more important to track changes in individual parameters over time rather than focusing on their exact values. 

## 6. Conclusions

We characterized the short- and long-term diastolic function trajectory after heart transplantation. Assessing LA phasic functions with speckle tracking echocardiography may provide meaningful parameters for identifying and monitoring diastolic dysfunction more accurately. Further, large-scale studies are warranted to test its added clinical value. 

## Figures and Tables

**Figure 1 diagnostics-14-01136-f001:**
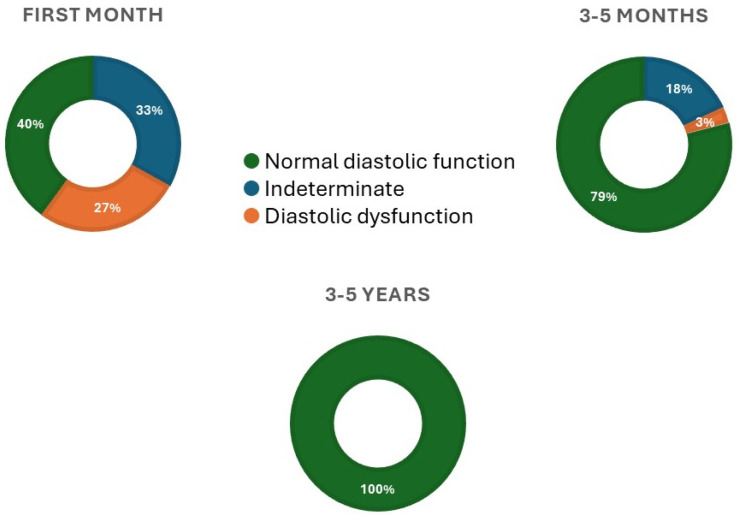
Distribution of diastolic function categories according to current echocardiographic guidelines at the three assessments after heart transplantation.

**Figure 2 diagnostics-14-01136-f002:**
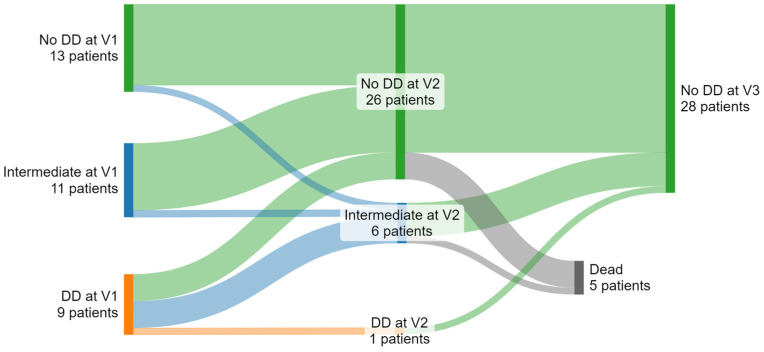
Sankey diagram to visualize reclassification from normal diastolic function (No DD), intermediate, and diastolic dysfunction (DD) over the three assessments (V1, V2, and V3).

**Figure 3 diagnostics-14-01136-f003:**
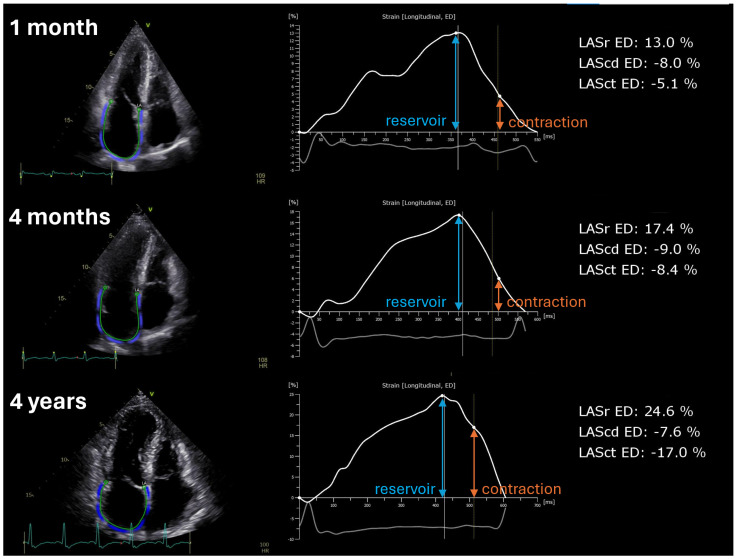
Left atrial deformation analysis of a representative patient—screenshots from the dedicated software’s user interface. (**Left**) panels: apical four-chamber view images with the overlaid region of interest. (**Right**) panels: volume–longitudinal strain curves with reservoir (LASr) and contraction (LASct) strains highlighted. The reservoir strain gradually increases over time. The contraction strain significantly improves from the second to third assessment.

**Table 1 diagnostics-14-01136-t001:** Basic demographic, hemodynamic, and clinical characteristics of the recipients and donors.

	HTX Patients at 1st Assessment (*n* = 33)
Age (years)	52 (46; 58)
Sex [female, *n* (%)]	9 (27%)
Height (cm)	176 (168; 182)
Weight (kg)	77 (69; 84)
BMI (kg/m^2^)	25.0 (22.0; 29.7)
BSA (m^2^)	1.94 (1.82; 2.06)
SBP (mmHg)	117 (100; 130)
DBP (mmHg)	75 (62; 80)
HR (1/min)	80 (76; 94)
Preoperative PVR (Wood units)	2.8 (1.9; 4.4)
Donor age (years)	46 (37; 55)
Donor sex [Female, *n* (%)]	11 (33%)
**Etiology**	
Nonischemic dilated cardiomyopathy [*n* (%)]	16 (49%)
Ischemic dilated cardiomyopathy [*n* (%)]	10 (30%)
Other, nonspecified [*n* (%)]	7 (21%)
**Comorbidities**	
Systemic hypertension [*n* (%)]	14 (42%)
Diabetes mellitus [*n* (%)]	8 (24%)
COPD/asthma [*n* (%)]	6 (18%)

BMI: body mass index; BSA: body surface area; COPD: chronic obstructive pulmonary disease; DBP: diastolic blood pressure; HR: heart rate; PVR: pulmonary vascular resistance; SBP: systolic blood pressure.

**Table 2 diagnostics-14-01136-t002:** Comparison between the 3 time points.

	1st Assessment	2nd Assessment	3rd Assessment	Overall *p*
Days since HTX	17 (13; 22)	115 (101; 125)	1492 (1377; 1560)	NA
NT-proBNP (pg/mL)	4560 (2421; 7205) ^ab^	711 (517; 1098) ^ac^	374 (177; 717) ^bc^	<0.001
GFR (mL/min/1.73 m^2^)	82 (55; 90)	69 (49; 84)	64 (47; 88)	0.268
**Echocardiography**				
LVIDd (mm)	45.0 (42.0; 48.0)	44.5 (42.0; 48.0)	46.0 (43.5; 48.0)	0.809
IVSd (mm)	12.0 (11.0; 14.0) ^b^	13.0 (11.0; 14.0) ^c^	11.0 (10.0; 13.0) ^bc^	0.004
PWd (mm)	11.0 (10.0; 11.0)	10.0 (9.0; 11.0)	10.0 (9.3; 12.0)	0.103
RWT (%)	46.8 (43.0; 52.4)	44.7 (38.1; 50.0)	45.5 (40.4; 52.5)	0.058
LVMi (g/m^2^)	102.4 (80.4; 116.1)	92.1 (82.0; 107.8)	89.0 (75.3; 102.8)	0.275
LVEDVi (mL/m^2^)	48.8 (39.0; 56.2)	47.7 (41.4; 52.5)	45.3 (41.5; 53.9)	0.446
LVESVi (mL/m^2^)	17.5 (13.3; 21.8)	17.4 (15.1; 20.3)	19.4 (16.9; 23.6)	0.331
LVEF (%)	64.0 (60.0; 66.0) ^b^	62.0 (60.0; 64.0) ^c^	57.0 (55.0; 60.0) ^bc^	<0.001
LV GLS (%)	−12.4 (−14.5; −10.5)	−12.6 (−15.3; −9.8)	−15.3 (−18.1; −12.2)	0.070
RVd (mm)	33.0 (29.5; 36.0)	32.0 (29.0; 34.5)	36.0 (33.0; 38.0)	0.058
TAPSE (mm)	11.0 (9.0; 14.0) ^a^	14.0 (12.0; 16.0) ^a^	13.0 (11.3; 15.0)	0.013
TR Vmax (m/s)	2.6 (2.5; 2.8)	2.6 (2.3; 2.7) ^c^	2.1 (1.7; 2.5) ^c^	<0.001
E (cm/s)	95.0 (84.0; 119.0) ^ab^	85.0 (69.5; 98.5) ^a^	84.0 (67.0; 94.0) ^b^	0.001
A (cm/s)	48.0 (42.0; 70.0)	47.0 (42.0; 65.0)	43.5 (35.0; 54.3)	0.786
E/A	2.03 (1.50; 2.35)	1.61 (1.48; 1.91)	1.76 (1.41; 2.22)	0.058
DT (ms)	163.0 (137.0; 182.5)	163.0 (150.0; 181.5)	171.0 (151.0; 200.0)	0.502
Mitral lateral s′ (cm/s)	10.0 (9.0; 11.0)	11.0 (9.0; 12.0)	10.0 (9.2; 11.5)	0.698
Mitral lateral e′ (cm/s)	11.0 (9.0; 13.5) ^ab^	15.0 (11.0; 16.0) ^a^	15.2 (13.2; 16.3) ^b^	0.001
Mitral lateral a′ (cm/s)	5.0 (4.0; 7.0) ^ab^	7.0 (5.0; 8.7) ^ac^	8.3 (6.9; 9.5) ^bc^	<0.001
Mitral medial s′ (cm/s)	8.0 (7.0; 9.0)	9.0 (8.0; 10.0)	8.7 (7.5; 9.3)	0.617
Mitral medial e′ (cm/s)	7.0 (6.0; 8.0) ^b^	8.0 (7.0; 9.0)	8.5 (7.7; 9.5) ^b^	0.002
Mitral medial a′ (cm/s)	6.3 (6.0; 8.0) ^b^	8.0 (5.7; 8.0)	8.8 (7.6; 9.6) ^b^	0.003
E/e′ average	10.5 (8.7; 12.5) ^ab^	7.9 (6.6; 9.1) ^a^	6.5 (5.8; 8.0) ^b^	0.002
LAVi (ml/m^2^)	53.1 (43.9; 67.9) ^b^	49.0 (37.5; 59.6) ^c^	33.6 (28.6; 40.6) ^bc^	<0.001
LA reservoir strain ED (%)	13.0 (7.0; 16.3) ^ab^	14.9 (13.1; 18.2) ^a^	20.0 (15.5; 21.8) ^b^	0.003
LA conduit strain ED (%)	−8.5 (−13.4; −4.35)	−9.3 (−13.2; −5.9)	−10.9 (−12.6; −9.0)	0.195
LA contraction strain ED (%)	−4.3 (−5.8; −1.2) ^b^	−4.5 (−8.9; −1.4) ^c^	−8.1 (−11.0; −5.8) ^bc^	0.015

^a^ significant between 1st and 2nd assessment; ^b^ significant between 1st and 3rd assessment; ^c^ significant between 2nd and 3rd assessment. A: mitral inflow velocity during atrial contractions; a′: peak late (atrial) diastolic annular velocity; DT: deceleration time; E: early diastolic mitral inflow velocity; e′: early diastolic annular velocity; EDVi: end-diastolic volume index; EF: ejection fraction; ESVi: end-systolic volume index; GFR: glomerular filtration rate; GLS: global longitudinal strain; HTX: heart transplantation; IVSd: interventricular septal thickness at end-diastole; LA: left atrial; LAVi: left atrial volume index; LV: left ventricle; LVIDd: LV internal diameter at end-diastole; Mi: mass index; PWd: posterior wall thickness at end-diastole; RVd: RV basal diameter; RWT: relative wall thickness; s′: systolic annular velocity; TAPSE: tricuspid annular plane systolic excursion; TR Vmax: tricuspid regurgitation peak velocity.

## Data Availability

The original contributions presented in the study are included in the article, further inquiries can be directed to the corresponding author.

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
