# Peer review of "Trajectory of Diastolic Function after Heart Transplantation as Assessed by Left Atrial Deformation Analysis"

_diagnostics, 2024, doi:10.3390/diagnostics14111136_

Round 1

Reviewer 1 Report

Comments and Suggestions for Authors

I am grateful to the editor for the opportunity to review the manuscript of Borbála Edvi et al. “Trajectory of diastolic function after heart transplantation as assessed by left atrial deformation analysis.” In this article, the authors studied the dynamics of left atrial deformation analysis in patients after heart transplantation for up to 5 years. The authors found that when assessing the diastolic function of the left ventricle using this method, it was normalized by the end of the observation period in all patients. It was previously shown that, along with indicators of deformation of the left and right ventricle (ref. 1-2, see below), the assessment of left atrial deformation analysis allows us to identify a cardiac rejection reaction. There is also evidence of improvement in left ventricular diastolic function after heart transplantation when assessed using traditional indicators. The authors assessed the dynamics of left atrial deformation analysis indicators, obtaining new scientific facts.

However, while reviewing the manuscript, I had the following comments and questions that I would like answers to from the authors:

1. I would like to clarify whether patients who showed cardiac rejection reactions were included in the study? As I understand it, such patients were re-examined after normalization of the biopsy data? Did such episodes of cardiac rejection affect the dynamics of left ventricular diastolic function?

2. I believe that simply references to echocardiography guidelines for diagnosing diastolic dysfunction are not enough. It is necessary to provide specific indicators and criteria for diagnosing diastolic function.

3. The authors studied a relatively small sample of patients. Accordingly, it is necessary to evaluate quantitative data for normality of distribution. If the distribution is different from normal, a different format for presenting data (median and quartiles), as well as other methods for assessing differences between indicators over time, will be necessary.

4. The authors state that “While LA strain is a promising new parameter for assessing cardiac allografts, the typical cut-off values ​​from literature data will not be suitable for HTX patients” (lines 182-183). This statement contradicts the authors' data, since the authors of the article, based on existing recommendations, are quite successful in assessing left ventricular diastolic function in patients. How can they explain this contradiction?

References:

1. Cruz CBBV, Hajjar LA, Bacal F, Lofrano-Alves MS, Lima MSM, Abduch MC, Vieira MLC, Chiang HP, Salviano JBC, da Silva Costa IBS, Fukushima JT, Sbano JCN, Mathias W Jr, Tsutsui JM. Usefulness of speckle tracking echocardiography and biomarkers for detecting acute cellular rejection after heart transplantation. Cardiovasc Ultrasound. 2021 Jan 9;19(1):6. doi: 10.1186/s12947-020-00235-w.

2. da Costa RCPL, Rodrigues ACT, Vieira MLC, Fischer CH, Monaco CG, Filho EBL, Bacal F, Caixeta A, Morhy SS. Evaluation of the myocardial deformation in the diagnosis of rejection after heart transplantation. Front Cardiovasc Med. 2022 Oct 13;9:991016. doi: 10.3389/fcvm.2022.991016.

Comments on the Quality of English Language

No comments

Author Response

Response: We would like to thank the expert Reviewer for his/her thorough assessment and great questions and suggestions. Accordingly, we have introduced several modifications to the text and results. Please see our point-by-point responses below.

I am grateful to the editor for the opportunity to review the manuscript of Borbála Edvi et al. “Trajectory of diastolic function after heart transplantation as assessed by left atrial deformation analysis.” In this article, the authors studied the dynamics of left atrial deformation analysis in patients after heart transplantation for up to 5 years. The authors found that when assessing the diastolic function of the left ventricle using this method, it was normalized by the end of the observation period in all patients. It was previously shown that, along with indicators of deformation of the left and right ventricle (ref. 1-2, see below), the assessment of left atrial deformation analysis allows us to identify a cardiac rejection reaction. There is also evidence of improvement in left ventricular diastolic function after heart transplantation when assessed using traditional indicators. The authors assessed the dynamics of left atrial deformation analysis indicators, obtaining new scientific facts.

Response: Thank you for thoroughly summarizing the novelty of our study!

However, while reviewing the manuscript, I had the following comments and questions that I would like answers to from the authors:

  1. I would like to clarify whether patients who showed cardiac rejection reactions were included in the study? As I understand it, such patients were re-examined after normalization of the biopsy data? Did such episodes of cardiac rejection affect the dynamics of left ventricular diastolic function?

Response: Thank you for raising this important point! The Reviewer understood well; we designed our protocol to only investigate such patients at each visit, who currently do not have ISHLT grade ≥II rejection diagnosed by concurrent endomyocardial biopsy. There were no such assessments; however, between the second and third visit, there were 9 patients with such episodes. All 9 patients had normal diastolic function at the second visit, and 8 of them also had normal diastolic function at the third visit. One patient with a rejection episode died before the third visit. Inspired by the Reviewer's question, we also looked at patients who died between the second and third visits. Four out of the five deceased patients had normal diastolic function, while one had intermediate. To showcase this finding and temporal dynamics of diastolic dysfunction grading, we created a Sankey diagram (Figure 2.) We also expanded the results:

"Four of the five deceased patients had normal diastolic function at the second assessment; the fifth was in the indeterminate category (Figure 2). All nine patients who had ISHLT grade ≥II rejection episodes had a normal diastolic function before the episode at the second assessment; eight patients remained in the normal function category after it, while one patient died before the third assessment."

2. I believe that simply references to echocardiography guidelines for diagnosing diastolic dysfunction are not enough. It is necessary to provide specific indicators and criteria for diagnosing diastolic function.

Response: The Reviewer is right. Accordingly, we have expanded the Methods as follows:

"Based on current guideline recommendations, we categorized each patient at each time point into normal diastolic function, indeterminate, and DD subgroups [3]. We used the following pathological cutoffs: average E/e′ > 14; septal e′ velocity < 7 cm/s or lateral e′ velocity <10 cm/s; tricuspid regurgitation peak velocity > 2.8 m/s; and LA volume index >34 ml/m2. Diastolic function is normal if more than half of the available variables do not meet the cutoff values for identifying DD. DD is present if more than half of the available parameters meet these cutoff values. The category is indeterminate if half of the parameters do not meet the cutoff values [3]."

3. The authors studied a relatively small sample of patients. Accordingly, it is necessary to evaluate quantitative data for normality of distribution. If the distribution is different from normal, a different format for presenting data (median and quartiles), as well as other methods for assessing differences between indicators over time, will be necessary.

Response: Thank you for highlighting this important statistical consideration! Accordingly, we have assessed normality of each variable using the Shapiro-Wilk test and temporal changes were investigated by repeated measures ANOVA in the case of normal distribution and Friedman test in the case of non-normal distribution. As the majority of our variables were of non-normal distribution, we decided to have all continuous data in the format of median and interquartile range (effected Table 1 and 2). There were minor changes in our results (we corrected the Results section accordingly) but no changes concerning LA strain metrics. 

The modified Methods section:

"Normal distribution of variables was tested using the Shapiro-Wilk test. As the majority of the variables were non-normally distributed, continuous data were presented as median and interquartile range (first quartile; third quartile), while categorical data were presented as counts and percentages (% of the total population). The Sankey diagram was constructed using SankeyMATIC (https://sankeymatic.com) to visualize the volume of reclassified patients over the 3 time points.  To compare variables across the 3 time points, repeated measures ANOVA was used, while time point pairs were compared with the least significant difference test in the case of normal distribution, while the Friedman test was used, followed by the Wilcoxon test with Bonferroni adjustment in the case of non-normal distribution. Spearman rank correlation test was used to assess the correlation between the variables. p values <0.05 were considered statistically significant."

4. The authors state that “While LA strain is a promising new parameter for assessing cardiac allografts, the typical cut-off values ​​from literature data will not be suitable for HTX patients” (lines 182-183). This statement contradicts the authors' data, since the authors of the article, based on existing recommendations, are quite successful in assessing left ventricular diastolic function in patients. How can they explain this contradiction?

Response: The Reviewer made a great point. HTX patients represent a unique population, and we believe that the conventional and new parameters referring to diastolic function should be handled cautiously. Thus, in this scenario, we think that neither LA reservoir strain assessment nor dysfunction grading by the current guideline can provide 100% certainty in diagnosing or rejecting elevated filling pressures. This should be tested against the gold standard assessment of filling pressures. So, while the Reviewer is right that both conventional and strain assessments point in the same direction (normalization of diastolic function), and there is no clue for inadequate adjudication, we should not conclude that we captured every single case of diastolic dysfunction.

References:

1. Cruz CBBV, Hajjar LA, Bacal F, Lofrano-Alves MS, Lima MSM, Abduch MC, Vieira MLC, Chiang HP, Salviano JBC, da Silva Costa IBS, Fukushima JT, Sbano JCN, Mathias W Jr, Tsutsui JM. Usefulness of speckle tracking echocardiography and biomarkers for detecting acute cellular rejection after heart transplantation. Cardiovasc Ultrasound. 2021 Jan 9;19(1):6. doi: 10.1186/s12947-020-00235-w.

2. da Costa RCPL, Rodrigues ACT, Vieira MLC, Fischer CH, Monaco CG, Filho EBL, Bacal F, Caixeta A, Morhy SS. Evaluation of the myocardial deformation in the diagnosis of rejection after heart transplantation. Front Cardiovasc Med. 2022 Oct 13;9:991016. doi: 10.3389/fcvm.2022.991016.

Response: Thank you for highlighting these relevant papers, we have included them among references after the corresponding sentence:

"Similarly to LV and RV deformation parameters, LA reservoir strain may have a role in detecting acute cellular rejections after HTX [6-8]."

Thank you once again for your thorough review and great suggestions! We hope you will find the paper now acceptable for publication.

Reviewer 2 Report

Comments and Suggestions for Authors

This brief report by Edvil et al characterizes the patterns of dyastolic dysfunction in heart transplant recipients. They comprehensively describe the evolution of echography parameters in their patients as time went by, not limited to dyastolic dysfunction and make a brief correlation to NT-pro BNP. They also offer high quality echo images.

Some minor comments are listed below.

Lines 50-51: could you ad some data about the most recent guidelines recommandations for DD dyagnosis?

Can you expand the echo- NT-pro BNP discussion?

Author Response

Response: We would like to thank the expert Reviewer for his/her thorough assessment and great questions and suggestions. Accordingly, we have introduced several modifications to the text. Please see our point-by-point responses below.

This brief report by Edvil et al characterizes the patterns of dyastolic dysfunction in heart transplant recipients. They comprehensively describe the evolution of echography parameters in their patients as time went by, not limited to dyastolic dysfunction and make a brief correlation to NT-pro BNP. They also offer high quality echo images.

Response: Thank you for your commending words!

Some minor comments are listed below.

Lines 50-51: could you ad some data about the most recent guidelines recommandations for DD dyagnosis?

Response: Thank you for the great suggestions, we have added the following to the Methods section:

"Based on current guideline recommendations, we categorized each patient at each time point into normal diastolic function, indeterminate, and DD subgroups [3]. We used the following pathological cutoffs: average E/e′ > 14; septal e′ velocity < 7 cm/s or lateral e′ velocity <10 cm/s; tricuspid regurgitation peak velocity > 2.8 m/s; and LA volume index >34 ml/m2. Diastolic function is normal if more than half of the available variables do not meet the cutoff values for identifying DD. DD is present if more than half of the available parameters meet these cutoff values. The category is indeterminate if half of the parameters do not meet the cutoff values [3]."

Can you expand the echo- NT-pro BNP discussion?

Response: Thank you for raising our attention to this important point! Accordingly, we have expanded the Discussion as follows:

"Reservoir strain improves early along with, for example, E/e’, and its absolute values and changes correlate with NT-proBNP. This latter is a remarkable finding, as this phenomenon may reflect that, similarly to NT-proBNP, reservoir strain can be a marker of „overall” cardiac health by integrating LV systolic and diastolic function, filling pressures, and structural remodeling. Despite the fact that the association of absolute NT-proBNP levels with acute cellular rejection is weak, temporal changes provide important clinical information [11]. The temporal changes of the neurohormone were also associated with the temporal changes in the LA reservoir strain, suggesting a pathophysiological link between the two biomarkers and justifying the findings of Rodriguez-Diego about the usefulness of reservoir strain in detecting rejection episodes [6]."

Thank you once again for your thorough review and great suggestions! We hope you will find the paper now acceptable for publication.

Round 2

Reviewer 1 Report

Comments and Suggestions for Authors

The authors of the article responded to my comments and made corrections to the text of the manuscript. I have no other comments.

Comments on the Quality of English Language

No comments